# New Advances in Using Virus-like Particles and Related Technologies for Eukaryotic Genome Editing Delivery

**DOI:** 10.3390/ijms23158750

**Published:** 2022-08-06

**Authors:** Pin Lyu, Baisong Lu

**Affiliations:** 1School of Physical Education, Hangzhou Normal University, Hangzhou 311121, China; 2Wake Forest Institute for Regenerative Medicine, Wake Forest University Health Sciences, Winston-Salem, NC 27157, USA

**Keywords:** virus-like particle (VLP), viral capsid, aptamer, aptamer-binding protein, genome editing, designer nuclease, delivery, RNA, ribonucleoprotein, ZFN, TALEN, CRISPR/Cas9

## Abstract

The designer nucleases, including Zinc Finger Nuclease (ZFN), Transcription Activator-Like Effector Nuclease (TALEN), and Clustered Regularly Interspaced Short Palindromic Repeats/CRISPR-associated (CRISPR/Cas), have been widely used for mechanistic studies, animal model generation, and gene therapy development. Clinical trials using designer nucleases to treat genetic diseases or cancers are showing promising results. Despite rapid progress, potential off-targets and host immune responses are challenges to be addressed for in vivo uses, especially in clinical applications. Short-term expression of the designer nucleases is necessary to reduce both risks. Currently, delivery methods enabling transient expression of designer nucleases are being pursued. Among these, virus-like particles as delivery vehicles for short-term designer nuclease expression have received much attention. This review will summarize recent developments in using virus-like particles (VLPs) for safe delivery of gene editing effectors to complement our last review on the same topic. First, we introduce some background information on how VLPs can be used for safe and efficient CRISPR/Cas9 delivery. Then, we summarize recently developed virus-like particles as genome editing vehicles. Finally, we discuss applications and future directions.

## 1. A Brief Introduction to Designer Endonucleases

Designer endonucleases enable scientists to edit target genomes and achieve gene knockout and DNA addition with unprecedented precision [1]. Early gene editing technologies achieved DNA editing through physical and chemical mutagenesis and homologous recombination. But these methods often fell short in efficiency, specificity, and practicality. The emergence of Zinc Finger Nuclease (ZFN) and Transcription Activator-Like Effector Nuclease (TALEN) greatly improved the precision and the practicality of gene editing. Nowadays, the most popular gene editing tool is the Clustered Regularly Interspaced Short Palindromic Repeats/CRISPR-associated (CRISPR/Cas) system. The development of these designer endonucleases enables researchers to make good use of non-homologous end-joining and homologous recombination for gene editing. However, there are still challenges to be overcome, such as off-target effects and immune responses [2,3]. One strategy to reduce these risks is transient delivery of the nucleases.

In the past several years, many groups have developed various lentivirus- or retrovirus-like particles (VLPs) for delivering mRNA, protein, or ribonucleoprotein of designer endonucleases to improve genome editing safety in mammalian cells and animals. Since our last review on VLPs as delivery tools for genome editing [4], many new developments have been reported [5,6,7,8,9,10,11,12,13,14,15,16]. In this review, we will complement our previous review [4] in light of these new discoveries.

Since many VLP systems used interactions between aptamer and aptamer-binding proteins (ABPs) to recruit RNA or protein into VLPs, we begin with background information about bacteriophage aptamers and ABPs. Building on the information on lentiviral capsid proteins in the previous review [4], our discussions on aptamer and ABP will help readers to better understand the mechanisms for using VLPs for safe CRISPR/Cas9 delivery. Following the summary of new developments in using VLPs to deliver designer nucleases in forms of protein, ribonucleoprotein (RNP), or RNA for genome editing, we introduce the applications of various types of VLPs in genome editing, with emphasis on in vivo applications.

Currently, the commonly used designer endonucleases include ZFN, TALEN and CRISPR/Cas endonucleases. ZFN is an artificially designed nucleic acid endonuclease that contains 3–6 Cys_2_-His_2_ zinc finger protein tandem for specific DNA recognition and a non-specific nucleic acid endonuclease from FokI (the carboxy-terminal 96 amino acid residues) for DNA cleavage [17]. Each Fok I monomer is linked to a zinc finger protein to form a ZFN to recognize, bind and cleave a specific site. When the 2 monomeric ZFNs bind DNA in the opposite directions and the 2 recognition sites are at the proper distance (6~8 bp), the 2 FokI domains interact to form an enzymatic active unit and cleave the target DNA to form a double-strand break. As a first-generation gene editing tool, ZFNs have the disadvantages of relatively low gene editing efficiency and high toxicity. TALEN is a second-generation gene editing tool, similarly consisting of DNA recognition domains and FokI nucleic acid endonuclease domains [18,19]. Compared with ZFN, TALEN has a simpler process for design, better gene editing efficiency, and lower toxicity. However, due to the presence of many repeats in the sequence coding for the DNA-binding domains, the resultant DNA tends to be unstable.

The CRISPR/Cas system, as the third-generation artificial endonuclease, has been available since August 2012 [20]. It has become widely used by scientists worldwide in various fields for its simple design, low cost, and high efficiency of gene editing in eukaryotic cells [21,22,23,24]. CRISPR/Cas is an adaptive immune system that recognizes and specifically degrades foreign invading nucleic acid sequences in bacteria and archaea [25]. According to the number of Cas gene effector proteins, CRISPR consists of 2 classes and 5 types, a total of 16 subtypes [26]. Type II Cas proteins are represented by CRISPR/Cas9 proteins, which require only one protein to perform cleavage. During long-term evolution, bacterial immune systems store sequences of viral genomes or plasmids in their own DNA in the form of spacer sequences. The spacer sequences are transcribed into CRISPR RNA (crRNA). Once an invading DNA with a sequence complementary to crRNA (the Protospacer) is detected, Cas protein can be directed to digest the target sequence and resist invasion. The CRISPR/Cas9 endonucleases need two components to be enzymatically active: Cas9 nuclease and single-guide RNA (sgRNA), a fusion RNA from crRNA and trans-activating CRISPR RNA (tracrRNA) [20,27]. Cas9 protein will form a complex with sgRNA as Cas9:sgRNA ribonucleoprotein. When the target sequence is completely complementary to crRNA, the Cas9:sgRNA complex is activated to cleave the target region and cause a double-strand break [28].

The advantages of the CRISPR/Cas system are efficient and specific cleavage of target sites, simultaneous editing at multiple locations, and easy preparation. With these advantages, CRISPR/Cas9 endonucleases can be applied to human gene therapy [29,30,31,32], new drug development, and other biomedical research fields. This new gene editing system has been used in research on a wide range of major human diseases, including cancer, infectious diseases, and genetic and autoimmune diseases, and it facilitates personalized clinical treatment. Many CRISPR-based gene therapy clinical trials have been registered on ClinicalTrials.gov, and some reported promising results [33,34,35,36]. In addition, the CRISPR/Cas system is also widely used in animal model generation [37], crop improvement [38], microbial genome editing [39], and gene expression regulation [40,41,42].

However, the potential of off-target effects and immune responses to the bacteria-derived Cas9 protein induce challenges in in vivo and clinical applications of the CRISPR/Cas system, especially when expressed in cells for the long-term [43,44,45,46]. These challenges hinder the progress of gene editing toward clinical treatments [2,47]. Developing delivery strategies featuring short-term designer nuclease expression is one way to resolve the challenges. Lenti- or retrovirus-like particles (VLPs) can be developed for efficient delivery of CRISPR/Cas effectors for short-term endonuclease expression.

## 2. Background Knowledge for Developing VLPs as Genome Editing Delivery Vehicles

An understanding of the capsids of lentiviruses and retroviruses is needed to learn the principles of VLPs. Please see our previous review [4] and others for more information on these vectors [48,49]. In many types of VLPs, the specific interactions between aptamers and ABPs were utilized to deliver protein or RNA by VLPs for genome editing. Here, we give a brief introduction to aptamers and ABPs used by various groups in developing VLPs.

### RNA Aptamers and Aptamer-Binding Proteins (ABPs)

The study of RNA or DNA aptamers started even before Andy Ellington first coined the term “aptamer” [50]. Aptamers are selective affinity reagents that can be used in a wide range of research, diagnostic, and therapeutic applications. They are usually 20~60 nucleotide single-stranded DNA or RNA molecules that can selectively bind to a specific target [51], including proteins, peptides, carbohydrates, small molecules, metal ions and even live cells. Here, we focus on RNA aptamers used in developing VLPs for genome editing delivery. At least four pairs of RNA aptamers/ABPs—MS2/MS2 coat protein (MCP) [52], PP7/PP7 coat protein (PCP) [53], com/Com [54], and BoxB/λ N22 peptide [55]—have been used in VLP-mediated RNA or RNP delivery (Table 1). Each aptamer specifically binds to its ABP with high affinity.

Aptamer MS2 and ABP MCP (MS2 coat protein) were found in bacteriophage MS2, one of a group of single-stranded RNA coliphages, including the bacteriophages MS2, R17, and f2. These bacteriophages contain only three genes, coding for synthetase, major coat protein, and maturation protein, respectively [56]. Aptamer MS2, adopting a stem–loop motif, is a 19 nt sequence found in the operon of synthetase (Figure 1). Specific binding of MCP to MS2 aptamer represses the translation of synthetase [52,57]. The interaction of MS2 and MCP was first used in studying cellular RNA localization in yeast [58] and subsequently used in dead Cas9 mediated gene regulation [59,60,61] and DNA labeling [62]. Recently, it has been used for VLP-mediated delivery of Cas9 mRNA or RNPs [5,52,63,64].

PP7 is a single-strand RNA bacteriophage of *Pseudomonas aeruginosa* that is distantly related to coliphage MS2 (Figure 1) [53]. Aptamer PP7 is a short sequence found in the translation initiation region of bacteriophage PP7 replicase gene; similar to aptamer MS2, it also adopts a stem–loop structure. Aptamer PP7 is specifically bound by PP7 coat protein (PCP) with an equilibrium dissociation constant of approximately 1 nm to repress the translation of synthetase [53]. Aptamer PP7 and PCP have been used in yeast [65] and mammalian cells [66] for RNA imaging.

Aptamer BoxB and ABP λ peptide N22 were found in bacteriophage λ (Figure 1). Peptide N22 is the amino terminal part of the λ phage N gene product and specifically binds to RNA hairpin BoxB in N protein-regulated genes to counteract transcription regulation [67]. An engineered λ N22 peptide showed greatly improved affinity to λ aptamer BoxB [55], and this aptamer/ABP pair was used to develop tools for genomic DNA labeling [62].

The RNA aptamer com and its interacting protein Com were found in the bacteriophage Mu (Figure 1) [68]. The sequence of the com aptamer was in the translational initiation region of *mom* gene. Binding Com protein to the com sequence stimulates the translation of *mom* gene [54]. The com/Com interaction has been used in dead Cas9-mediated gene regulation [60] and DNA labeling [62].

## 3. Recently Developed VLPs as Safe Genome Editing Delivery Vehicles

Since our last review of various types of VLPs developed for genome editing delivery [4], many new types of VLPs have been developed for different purposes. These newly developed VLPs and those introduced in our last review are listed in Table 2 (RNP delivery) and Table 3 (RNA delivery). In the following subsections, we will only introduce VLPs reported after the publication of our last review. Readers are referred to our previous review and the original papers listed in the Tables for more information on prior VLPs. We will introduce these recently reported VLPs for RNP delivery (Table 2) and mRNA delivery and mRNA/sgRNA co-delivery (Table 3).

### 3.1. Using VLPs to Deliver Proteins or Ribonucleoproteinss

Before being developed as genome editing delivery tools, VLPs were used for protein delivery for the purposes of viral-particle tracking and cell elimination [69,70,71]. The need for transient expression of genome editing effectors promoted the use of VLPs to deliver designer nuclease proteins or ribonucleoproteins (RNPs). Two strategies were used to package cargoes into the particles: the fusion strategy and the aptamer/ABP interaction strategy (Table 2). The fusion strategy was first used by Cai et al. to deliver ZFN and TALEN proteins by VLPs [72]. It was subsequently used by Choi et al. for packaging and delivering Cas9 RNPs (the Cas9:sgRNA complex) [73]. The fusion strategy was first used by Lyu et al. to deliver Cas9 RNPs by VLPs [63]. Here, we also include work using exosomes as delivery vehicles for genome editing [5] since these particles are similar in size to VLPs and also have membranes obtained from the producing cells. Figure 2 illustrates the packaging strategies of the recently reported VLPs.

**Table 2 ijms-23-08750-t002:** VLP-mediated nuclease protein and RNP delivery.

	Reference	Mechanism for Nuclease Recruitment	Capsid Type	Editing Effectors Delivered	Experiment Stage	Gene/Tissue
The fusion strategy	Cai et al. [72]	Fusing editing effector to the N-terminus of Gag	LV	ZFN and TALEN	In vitro	* GFP/CCR5/AAVS1 * — cell
	Choi et al. [73]	Fusing Cas9 protein to the N-terminus of Gag	LV	SpCas9	In vitro	* LTR/CD4/CCR5 * — cell
	Mangeot et al. [74]	Fusing Cas9 to the C-terminus of MLV Gag	MLV	SpCas9	In vivo	* MYD88/DDX3/GFP/Hpd/Fto/Tyr/LoxP * — cell
	Gee et al. [75]	Fusing FKBP12 to Gag, fusing FRB to SpCas9. FKBP12/AP21967/FRB interaction brings SpCas9 to Gag	LV	SpCas9	In vivo	* DMD * — cell and mouse
	Indikova et al. [76]	Fusing Cas9 to the C-terminus of Vpr	LV	SpCas9	In vitro	* GFP/EMX1/FANCF/HEKs1/HEKs3 * — cell
	Hamilton et al. [10]	Fusing Cas9 to N-terminal Gag structural protein	LV	SpCas9	In vitro	*B2M/TRAC*—cell
	Banskota et al. [12]	Fusing ABE8e to C-terminus of Gag	MLV	SpCas9 and ABE	In vivo	*BCL11A/COL7A1*—cell *Pcsk 9/Dnmt 1*—mouse
The Aptamer/ABP Strategy	Lyu et al. [63,77]	Forming a three-component complex: NC-Com/aptamer-sgRNA/Cas9 protein	LV	SpCas9 and ABE	In vitro	* IL2RG/HBB—cell *
	Lu et al. [78]			SaCas9		
	Yao et al. [5]	Forming a three-component complex: CD63-Com/aptamer-sgRNA/Cas9 protein	Exosome	SaCas9 SpCas9 ABE	In vivo	*DMD/IL2RG/HBB*—cell *DMD*—mouse

Highlighted papers were covered in our previous review [4].

#### 3.1.1. VLP-Mediated Nuclease Delivery Using the Fusion Strategy

##### VLPs for Co-Delivery of Cas9 RNPs and LV Genomic RNA

Recently, VLPs were harnessed to deliver Cas9 RNPs and DNA templates for both targeted genetic disruption and stable gene addition [10]. In this strategy, Cas9 was fused to the C-terminus of the HIV Gag protein with cleavage linker sequences. The fusion protein will be incorporated into the VLPs formed with the help of unmodified Gag and Pol products. In addition to Cas9 RNPs, lentiviral genomic RNA will also be packaged via the interaction between NC protein and the packaging signal in the lentiviral genomic RNA. Sequences in the genomic RNA can be reverse-transcribed into DNA and integrated into the host genome for long-term gene expression or serve as DNA templates for homologous recombination if the VLPs are integration-defective (with an inactivated integrase). These VLPs could be especially useful in creating chimeric antigen receptor (CAR) T cells with therapeutically relevant gene disruption (e.g., B2M or TRAC). Indeed, pseudotyping the particles with different envelope proteins shifts the tropism of the particles, when transducing T cells.

##### Recent Applications of Previously Reported VLPs

Recently, there are also new applications of VLPs originally reported by Mangeot et al. [74]. Vindry et al. used this system to prepare VLPs to deliver Cas9 RNPs targeting the selenocysteine-tRNA[Ser]^Sec^ gene in multiple cell types [79]. They observed >80% genome editing efficiency in multiple cell types tested, confirming the efficiency of the VLPs. Gutierrez-Guerrero et al. explored the co-pseudotyping of the VLPs developed by Mangeot et al. [74] with baboon endogenous virus (BaEV) envelope glycoprotein and VSV-G, and they achieved relatively high gene editing efficiency (>25%) in human B, T cells and hematopoietic stem and progenitor cells [9]. Thus, work from multiple groups shows that VLPs can be pseudotyped with different envelope proteins to improve gene editing efficiency in peculiar cell types.

##### VLPs for Delivering Base Editors

Base editors (BEs), which mediate targeted single-nucleotide conversions, can minimize the risks of generating double-strand breaks compared with artificial endonucleases. They were commonly delivered by plasmid DNA, AAV (needs to be split into two AAV vectors due to the large size of adenine base editors), and LV. Previously Lyu et al. developed VLPs to deliver an adenine base editor (ABE), using the interactions between aptamer and ABP [77]. They found that VLP-delivered ABEs did not show sgRNA-independent RNA off-target effects. Recently, Banskota et al. used a different strategy to deliver base editors by VLPs and called these VLPs engineered DNA-free virus-like particles (eVLP) [12]. They fused ABE8e to the C-terminus of the Friend murine leukemia virus (FMLV) gag polyprotein via a linker peptide to allow for cleavage by the FMLV protease upon particle maturation. They optimized the cleavage site to improve cargo release, added a nucleus export signal to improve cytoplasmic availability of the Gag-cargo for packaging, and adjusted the ratio of Gag-cargo and unmodified Gag to further improve particle generation and base editing activities. These efforts resulted in high efficiency of base editing. The authors successfully used these VLPs for in vivo base editing in mice. Systemically delivered eVLPs efficiently edited the gene *Pcsk9* in mouse liver and resulted in a 78% reduction in serum Pcsk9 protein levels [12].

#### 3.1.2. Delivering Cas9 and Base Editor RNPs by Exosomes Using the ABP/Aptamer Interaction Strategy

Previously, Lyu et al. and Lu et al. used the specific interactions between aptamers and ABPs to package and deliver Cas9 and ABE RNPs by VLPs [63,78,80]. Recently, they adapted the strategy to package and deliver Cas9 and ABE RNPs by extracellular vesicles (exosomes) [5], which is an alternative delivery vehicle to VLPs. Although exosomes are different from VLPs in that exosomes do not have virus-derived capsids, exosomes do have similarities with VLPs in size (~100 nm) and membranes. The authors fused ABP Com to the N- and C-termini of CD63, a membrane protein abundant in exosomes [81,82], and showed that SaCas9 RNPs, SpCas9 RNPs, and ABE RNPs can all be enriched and delivered by exosomes. Exosome-delivered RNPs achieved moderate gene edit efficiencies in HEK-293T cells, MDA-MB-231 cells, and human muscle progenitor cells. Compared with delivering RNPs by VLPs, delivering RNPs by exosomes usually needs more HEK-293T cells to generate particles for comparable genome editing efficiencies on the same target. In the fusion strategy to use VLPs for RNP delivery, both VLPs and exosomes are most likely produced. However, since there is a mechanism to enrich RNPs into the VLPs but not exosomes, the vast majority of the genome editing activities are from the VLPs.

### 3.2. VLPs for Delivering Cas9 mRNAs

In the past, VLPs for mRNA delivery were developed from viral capsids [15,64,83,84,85]. Recently, endogenous retrotransposon proteins were used to develop VLPs for the same purpose [11]. Segel et al. screened Gag homologs in mouse and human genomes and found that PEG10 could form capsids with RNA packaging capability. They developed a selective endogenous encapsulation for cellular delivery (SEND) system consisting of three modules: the capsid assembling protein PEG10, the cargo mRNA bearing a cis element interacting with PEG10, and the endogenous fusogenic protein SYNA (to replace commonly used VSV-G) for cell entry and escape from the endosome system. The SEND system can efficiently package and deliver SpCas9 mRNA bearing a PEG10-interacting cis element (Figure 3A). Consistent with previous observations [15,63,84], sgRNA alone could not be functionally delivered by these VLPs. Surprisingly, unmodified sgRNA could be functionally delivered by SEND in the presence of SpCas9 mRNA bearing a PEG10-interacting cis element, but the mechanism behind this delivery is unknown. Future in vivo applications of this endogenous VLP system in genome editing will likely be forthcoming.

### 3.3. VLPs for Co-Delivery of Cas9 mRNA and sgRNA

Several studies have found that sgRNA packaged alone could not be functionally delivered by VLPs [11,15,63,84], possibly because sgRNA is very unstable in cells unless complexed with Cas9 protein [86]. Accordingly, most mRNA-delivering VLPs do not co-deliver sgRNA, except the recently reported endogenous VLP (SEND) system [11], but its mechanism for sgRNA delivery is unclear.

Two different strategies were used to develop VLPs for co-delivering Cas9 mRNA and sgRNA (Table 3). In the first strategy, RNA aptamers were added to Cas9 mRNA and sgRNA, and the aptamer/ABP interactions were used to co-package Cas9 mRNA and sgRNA in VLPs (Figure 3B) [14,15,16]. In the second strategy, Cas9 mRNA was packaged via aptamer/ABP interactions by adding RNA aptamers in the Cas9 mRNA 3′ untranslated region (UTR), whereas the sgRNA expression cassette was packaged as part of the lentiviral genomic RNA via the interactions between nucleocapsid (NC) protein and lentiviral genomic RNA packaging signal ψ (Figure 4). The sgRNA expression cassette was then reverse-transcribed into DNA to mediate sgRNA expression [6,7,13].

**Table 3 ijms-23-08750-t003:** Virus-like particle (VLP) mediated RNA delivery.

	Reference	Virus Type	Capsid Modification	RNA Package	Copy Number
mRNA	Mock et al. [83]	LV	Not modified	TALEN mRNA	2 copies
	Prel et al. [85]	LV	MCP replaced the second zinc finger domain of NC	SpCas9 mRNA	~6 copies
	Lu et al. [64]	LV	MCP inserted after the second zinc finger domain of NC	SaCas9 mRNA	50~100 copies
	Lindel et al. [84]	Foamy Viruses	Not modified	SpCas9 mRNA	60 copies
mRNA or mRNA & sgRNA	Segel et al. [11]	Endogenous retrotransposon	No modification of endogenous Gag homolog	SpCas9 mRNA (and sgRNA)	Not available
mRNA & sgRNA	Knopp et al. [14]	Murine Leukemia Virus or Rous sarcoma virus	Two copies of MCP replaced NC	SpCas9 mRNA and sgRNA	Not available
	Baron et al. [15]				
	Mianné et al. [16]	LV	MCP inserted the N terminus of CA and PCP replaced ZNF2	SpCas9 mRNA and sgRNA	1.43 copies
hybrid mRNA & sgRNA	Ling et al. [6]	LV	MCP inserted the N terminus of Gag	SpCas9 mRNA and sgRNA	3~4 copies
	Yin et al. [7]				
	Yadav et al. [13]	LV	MCP inserted after the second zinc finger domain of NC	SaCas9 mRNA and sgRNA	Not available

Highlighted papers were covered in our previous review [4].

#### 3.3.1. Adding Aptamers to Cas9 mRNA and sgRNA for Co-Delivery

Knopp et al. originally used VLPs developed from gammaretroviral murine leukemia virus to package Cas9 mRNA or co-package Cas9 mRNA and sgRNA [15]. Recently the same group used VLPs developed from alpharetroviral Rous sarcoma virus for improved efficiency [14]. In both studies, the authors replaced the nucleocapsid protein within Gag protein with 2 copies of MCP and added 2 copies of MS2 aptamer in the 3′ UTR of SpCas9 mRNA, and in various positions of sgRNA. The idea was to recruit Cas9 mRNA and sgRNA into the VLPs via MS2/MCP interactions. These VLPs successfully created DNA mutations in various murine and human cell lines, including human T cells, primary human fibroblasts, and cord-blood-derived CD34^+^ stem and progenitor cells [14,15].

The RNA-delivery VLP system originally reported by Prel et al. [85] was also recently modified for co-delivering Cas9 mRNA and sgRNA to human induced pluripotent stem cells [16]. In this method, two ABPs, MCP and PCP, were inserted into matrix protein (MA) and nucleocapsid protein, respectively; and two aptamers, PP7 and MS2, were inserted into sgRNA and Cas9 mRNA, respectively. The idea was to package sgRNA via PP7/PCP interactions and package Cas9 mRNA via MS2/MCP interactions. The authors observed efficient genome editing by this co-delivery method.

In the studies described above using aptamer/ABP interactions to co-deliver Cas9 mRNA and sgRNA [14,15,16], delivering Cas9 RNPs rather than or in addition to Cas9 mRNA/sgRNA cannot be ruled out. Several lines of evidence suggest that in these VLPs, the packaged Cas9 RNPs could be the main contributor to gene editing activities. First, several reports showed that sgRNA alone could not be functionally delivered by VLPs [11,15,63,84], and it is difficult to explain how sgRNA could be stabilized in the presence of Cas9 mRNA if Cas9 protein is not involved. Second, when co-expressing aptamer-negative Cas9 mRNA with aptamer-positive sgRNA in VLP producing cells, the aptamer/ABP interaction was enough to recruit Cas9 RNPs into VLPs [63]. This was because in VLP-producing cells, Cas9 proteins were translated from Cas9 mRNAs, and the former have an intrinsic affinity with sgRNAs that are recruited into VLPs via aptamer/ABP interactions. Indeed, Mianne et al. observed that VLPs prepared with aptamer-negative Cas9 mRNA and aptamer-positive sgRNA had 80% of the gene editing activity of VLPs prepared with Cas9 mRNA and sgRNA that were both aptamer-positive [16]. This finding suggests that approximately 80% of the genome editing activity could be contributed by packaged Cas9 RNPs. Thus, in these VLPs, a mixture of Cas9 RNPs, Cas9 mRNAs, and sgRNAs could have been delivered, and Cas9 RNPs might have contributed to the majority of the genome editing activities.

#### 3.3.2. VLP and LV Hybrid Particles for Cas9 mRNA and sgRNA Co-Delivery

Two groups used a different strategy to co-deliver Cas9 mRNA and sgRNA by VLPs [6,7,13] (Figure 4). Ling et al. and Yin et al., both from the Cai group, fused MCP to the N-terminus of Gag and added multiple copies of the MS2 aptamer to the 3′ UTR of SpCas9 mRNA [6,7]. The SpCas9 mRNA was packaged by interactions between MCP and MS2. Simultaneous transfection of sgRNA-expressing lentiviral transfer plasmid DNA into the VLP-producing cells produced hybrid VLP/LV particles containing SpCas9 mRNA and sgRNA-expressing LV genomic RNA. Since the integrase bears a D64V mutation, the sgRNA-expression cassette will not be integrated. In one study, the authors used these hybrid particles to target vascular endothelial growth factor A (*Vegfa*) to suppress aberrant development of blood vessels in a mouse model of wet age-related macular degeneration [6]. In another study, the authors used these hybrid particles to target herpes simplex virus type 1 (HSV-1) DNA and then remove integrated viral DNA from cells in the eye, successfully blocking herpetic stromal keratitis [7]. In both studies, delivering the particles to the eyes of mice produced evident therapeutic effects [6,7].

Yadav et al. used a similar strategy to co-deliver Cas9 mRNA and sgRNA [13] but a different way to incorporate ABP into lentiviral capsids. They inserted ABP after the second zinc finger domain of the NC protein, which preserved capsid assembly efficiency [63,64,78]. They compared the four pairs of aptamer/ABPs listed in Table 1 for VLP-mediated RNA delivery and found that com aptamer and Com ABP was the most efficient pair for this purpose. They similarly expressed sgRNA from a sgRNA-expressing cassette contained in the LV genomic RNA. Co-delivery of Cas9 mRNA and sgRNA by these hybrid VLPs edited the genome more efficiently than delivering RNPs by VLPs. This observation suggests the value of co-delivering Cas9 mRNA and sgRNA.

## 4. Applications of VLPs in Genome Editing

VLPs as delivery vehicles for genome editing are still in preclinical development. However, successful applications have been reported in clinically relevant primary cells and disease animal models. Below, we summarize these applications.

### 4.1. In Vitro Applications in Clinically Relevant Blood Cells

Currently, genome editing in human blood cells relies on electroporation of Cas9 RNPs into them, which may affect cell viability. VLPs can deliver Cas9 RNPs into human hematopoietic stem cells to disrupt the *Myd88* gene [74], whose mutation causes immunodeficiency. More recently, VLPs (‘Nanoblades’) were co-pseudotyped with VSV-G and baboon endogenous virus envelope glycoproteins to achieve better genome editing efficacy in human B, T, and CD34^+^ hematopoietic stem and progenitor cells; they showed little effects on cell viability or differentiation [9]. In another study, VLPs efficiently knocked out genes in primary CD4^+^ T cells [76].

Currently, generating CAR-expressing T cells with gene disruption depends on transduction of retroviral vectors and electroporation of Cas9 RNPs. Hybrid VLPs containing Cas9 RNPs and CAR-expressing lentiviral genomic RNAs have been used for simultaneous CAR insertion and gene (for example *B2M* or *TRAC*) disruption [10]. Pseudotyping the VLPs with the HIV-1 envelope glycoprotein increased the transduction of CD4^+^ T cells compared to CD8^+^ T cells (53.20% versus 2.51% gene disruption efficiency).

### 4.2. In Vivo Applications of VLPs in Mice

VLPs have been used to deliver Cas9 RNPs into mouse zygotes to disrupt the tyrosinase (*Tyr*) gene, and insertions and deletions were observed in 16 of 40 blastocysts injected [74]. In addition, the authors delivered VLPs of *loxp*-targeting Cas9 RNPs into the perivitelline space of R26R-EYFP embryos, and successfully removed the “lox-stop-lox” cassette to enable EYFP expression in mice. Finally, they delivered VLPs with RNPs targeting the hydroxyphenylpyruvate dioxygenase (*Hpd*) gene into *Fah*^-/-^ mice via retro-orbital injection and obtained gene disruption in 7% to 13% of alleles [74].

In another study, *Dnmt1*-targeting ABE RNPs were delivered by VLPs into the central nervous system of neonatal mice via cerebroventricular injections. The authors observed over 50% base editing efficiency in VLP-transduced cells [12]. By delivering *Pcsk9*-targeting ABE RNPs into adult mice via retro-orbital injections, they achieved 63% editing in the liver and a 78% reduction in serum Pcsk9 protein levels. Finally, by delivering *Rpe65*-targeting ABE RNPs into the eyes of *rd12* mice with a silence mutation that causes loss of vision, the authors observed 11.5% to 21% base editing and rescued visual function [12].

Cas9 mRNA and sgRNA-delivering VLP/LV hybrid particles were also used in vivo [6,7]. In both studies, the particles were delivered to the eyes of disease mouse models to disrupt the disease-causing DNA: *Vegfa*, which resulted in abnormal blood vessel formation in one study [6] and HSV-1 viral DNA causing infection in the eye in the other study [7]. Phenotypic improvements were observed after VLP local delivery.

Two types of Cas9 RNP-delivery particles, NanoMEDIC [75] and engineered exosomes [5], have been injected into muscle tissues of Duchenne muscular dystrophy (DMD) mouse models and achieved detectable levels of genome editing. Although local muscle delivery may not be an idea for DMD treatment, these studies demonstrated that these particles can be used for delivery into the muscle tissue.

In summary, VLPs have successfully delivered Cas9 RNPs, ABE RNPs, or Cas9 mRNA/sgRNA into mouse eyes, central nervous system, and muscle via local injections and into mouse livers via systemic injections [5,6,7,12,74]. Combined with tissue-specific pseudotypes, VLPs may be used for CRISPR/Cas9 delivery for genome editing in other organs.

## 5. Summary

This review presents various VLPs developed for delivering RNAs or RNPs for genome editing. The common feature for all of the VLPs is the short-term expression of the effectors. When the format of cargoes is considered, all VLPs can be classified as RNA VLPs and RNP VLPs. The two types of VLPs each have their own advantages and disadvantages: RNP VLPs are all needed in the particles for genome editing, whereas some RNA VLPs (except for those co-delivering Cas9 mRNA and sgRNA) need sgRNAs to be provided separately. The number of effectors packaged in RNP VLPs is pre-determined and controllable, whereas the number of effectors expressed from RNA VLPs may reach a much higher level during a short period of time. This high level of effectors is not desirable when off-targets are concerned, but it could be desirable when CRISPR/Cas9 cleavage is inhibited by heterochromatin [87,88,89] and more effectors are needed to achieve desirable genome editing efficiency [90].

RNPs are packaged into capsids via two different mechanisms: the fusion mechanism and the aptamer/ABP interaction mechanism. With the fusion mechanism, fusing a large protein to Gag may impair capsid assembly efficiency and result in effector degradation by proteinase. With the aptamer/ABP interaction mechanism, inserting a small ABP into the NC protein of Gag has little effect on capsid assembly, whereas adding aptamers in sgRNA may impair genome editing activity with some target sequences. In addition, the release of RNPs from the ABPs cannot be controlled. It is hard to simply claim that one method is superior to another. The users have to decide which method best meets their specific needs.

Although VLPs have not been tested in clinical trials for genome editing, they have been successfully used in vitro in clinically relevant human cells and in vivo in disease mouse models. We hope that in the near future VLPs are tested as delivery vehicles for genome editing in clinical applications to benefit patients.

One common issue in using VLP as a genome editing delivery vehicle is the dependence of VSV-G or other viral envelope proteins to facilitate cell entry and endosomal escape. VSV-G and other viral envelopes could be toxic at high concentrations [91]. In addition, pre-existing immune responses to VSV-G or other viral envelope proteins may decrease in vivo genome editing efficiency. Recently it was found that mouse SYNA protein showed similar activities as VSV-G when working with retrotransposon derived Gag-like proteins [11]. It would be interesting to determine whether SYNA can replace VSV-G to facilitate cell entry and endosomal escape of VLPs. Using nonviral proteins to facilitate VLP cell entry and endosomal escape will improve safety and efficiency.

Finally, we would like to take this opportunity to encourage authors to try their best to include “virus-like particles (VLPs)” as one of the keywords while reporting related findings. Currently over twenty types of VLPs have been developed as genome editing delivery vehicles (see Table 2 and Table 3). Many authors gave a unique term to the VLPs they developed. If there is not a common term to refer to these different types of particles, unnecessary confusion could be caused. We suggest that authors try their best to use “VLP” (short form for “virus-like particles”) for general particles. “RNP VLP”, “mRNA VLP”, and “mRNA/sgRNA VLP” can be used to specify VLPs packaged with RNPs, mRNA, or mRNA plus sgRNA, respectively. We believe that unified terms will be helpful to the development of this field.

## Figures and Tables

**Figure 1 ijms-23-08750-f001:**
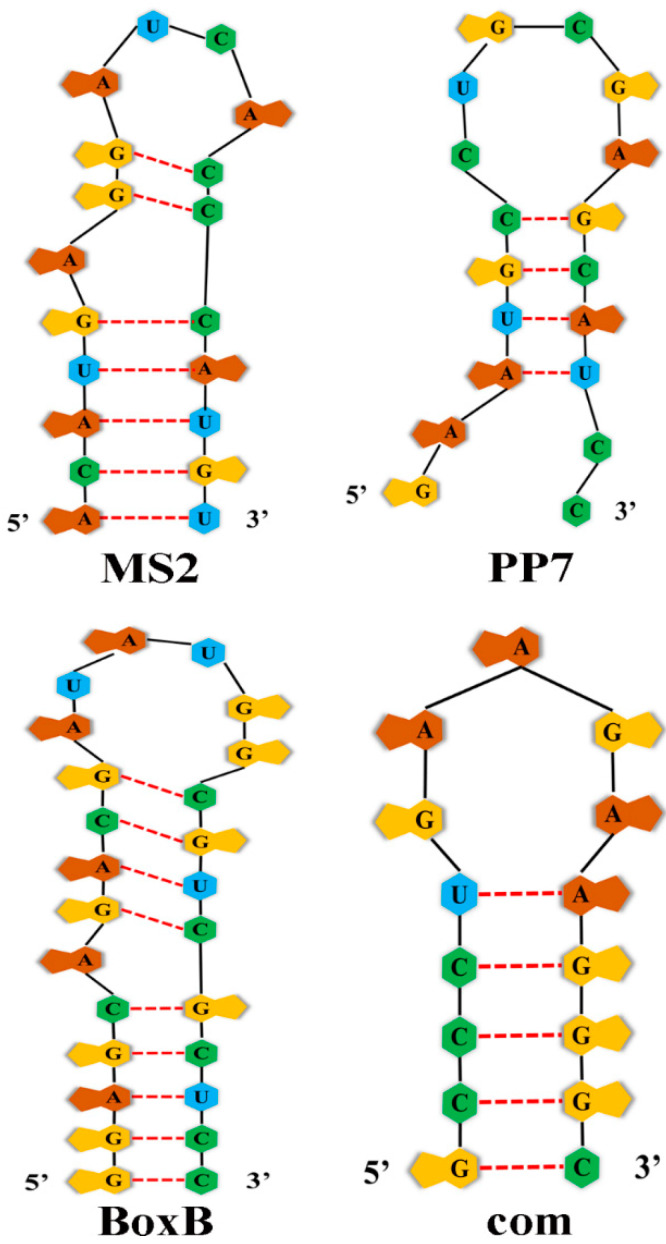
Structure of four RNA aptamers.

**Figure 2 ijms-23-08750-f002:**
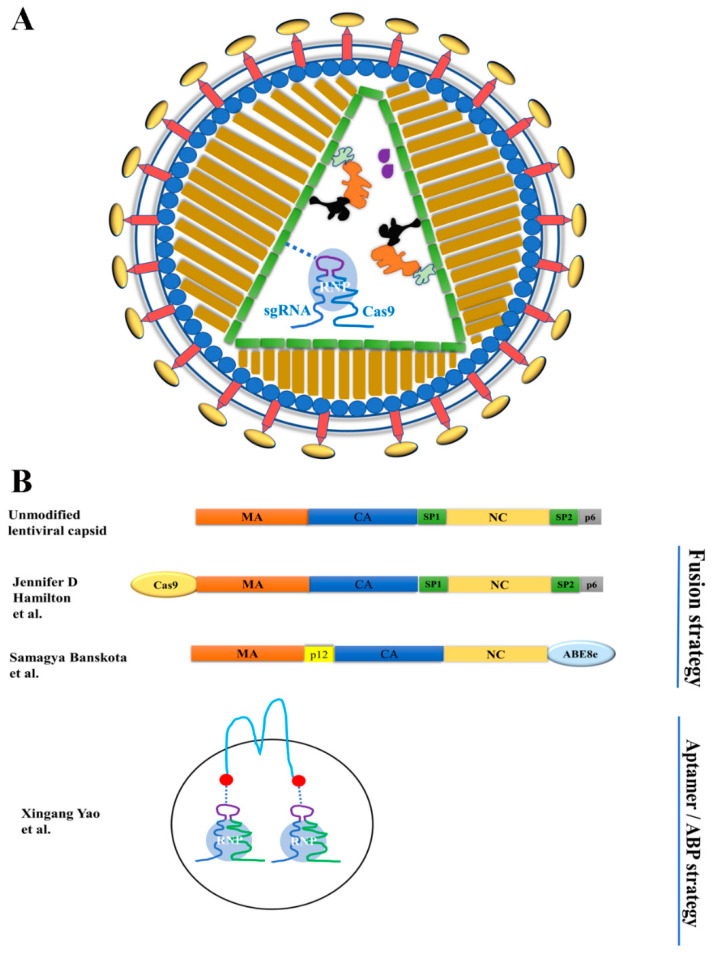
Strategies for modifying the Gag protein for designer nuclease RNP delivery by VLPs. (**A**) Diagram illustrating protein- or RNP-delivering VLPs. The RNA, if present, does not contain a long terminal repeat, so reverse transcription cannot happen. (**B**) Strategy for packaging proteins and RNPs into extracellular vesicles. A dashed line indicates non-covalent interactions. The VLPs and exosomes were reported in references [5], [10] and [12] respectively.

**Figure 3 ijms-23-08750-f003:**
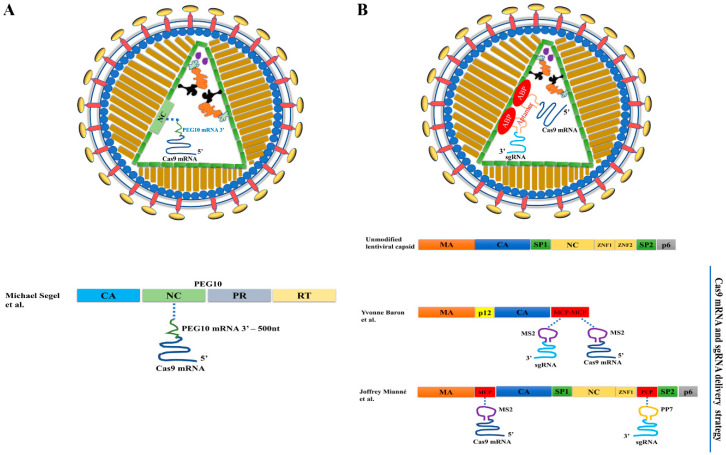
Using VLPs to deliver mRNA for genome editing. (**A**) Viral capsid-derived RNA-delivering VLPs. Aptamers are added in Cas9 mRNA and sgRNA so that they can be packaged via aptamer/ABP interactions. The RNAs do not contain a long terminal repeat, so reverse transcription cannot happen. (**B**) Mammalian transposon protein-derived VLPs for RNA delivery. The VLPs were reported in references [11], [15] and [16] respectively.

**Figure 4 ijms-23-08750-f004:**
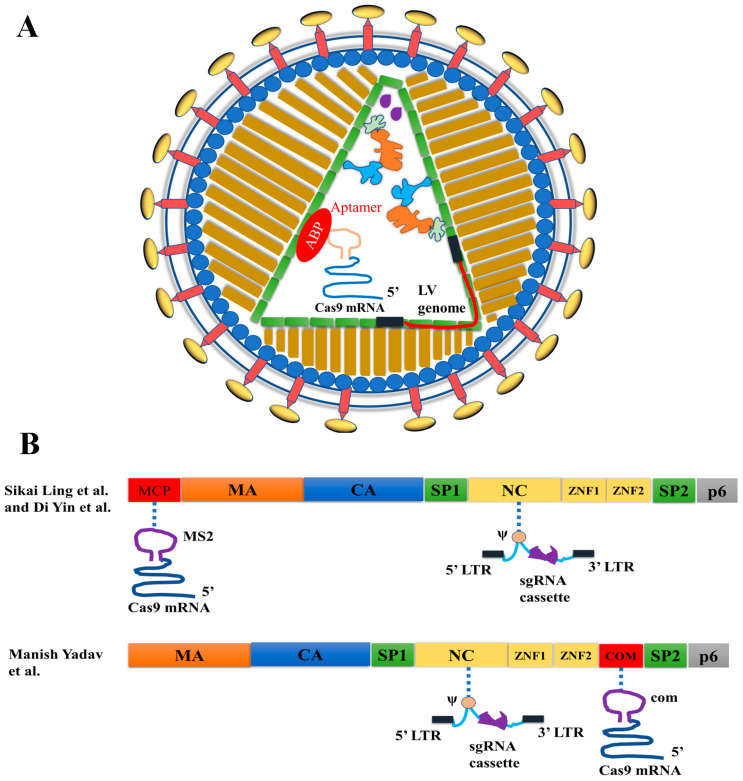
Strategies for modifying the Gag protein for mRNA and sgRNA delivery by VLP/LV hybrid particles. (**A**) Diagram illustrating mRNA- and sgRNA-co-delivery of VLP/LV hybrid particles. The mRNAs can only serve as the templates for translation since they do not contain viral sequences necessary for reverse transcription, so reverse transcription cannot happen. (**B**) Different approaches to Gag modification for Cas9-mRNA and sgRNA-expression cassette co-delivery. The VLPs were reported in references [6], [7] and [13] respectively.

**Table 1 ijms-23-08750-t001:** Aptamer and aptamer-binding protein (ABP) used for VLP-mediated RNA/RNP delivery.

Aptamer	Aptamer Sequence (5′ to 3′)	ABP	Size of ABP	Reference
MS2	ACAUGAGGAUCACCCAUGU	MCP	117 AA	[52]
PP7	GAAUGCCUGCGAGCAUCC	PCP	121 AA	[53]
BoxB	GGAGCAGACGAUAUGGCGUCGCUCC	λ N22	22 AA	[55]
com	GCCCUGAAGAAGGGC	Com	62 AA	[54]

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
