# Peer review of "New Advances in Using Virus-like Particles and Related Technologies for Eukaryotic Genome Editing Delivery"

_ijms, 2022, doi:10.3390/ijms23158750_

Round 1
Reviewer 1 Report
Pin Lyu and Baisong Lu present a quality and well-written review manuscript focused on new advances in using virus-like particles and related technologies for eukaryotic genome editing delivery.
Authors summarize recent developments in using virus-like particles for safe delivery of gene editing effectors. First they introduce some background information for easy understanding of how virus-like particles can be used for safe and efficient CRISPR/Cas9 delivery. Then they summarize recently developed virus-like particles as genome editing vehicles. Finally, they discuss their applications and future directions.
Authors cover such aspects as background knowledge for developing virus-like particles as genome editing delivery vehicle; recently developed virus-like particles as safe genome editing delivery vehicles; applications of virus-like particles in genome editing.
Finally, authors conclude that although virus-like particles have not been tested in clinical trials for genome editing, they have been successfully used in vitro in clinically relevant human cells and in vivo in disease mouse models. Authors hope that in the near future virus-like particles will become delivery vehicles for genome editing in clinical applications to benefit patients.
Overall, the manuscript is highly valuable for the scientific community and should be accepted for publication after the corrections are made.
==============================
Other comments:
1) Please check for typos throughout the manuscript.
2) With regards to gene editing authors are kindly encouraged to cite the following article that describes the use of CRISPR/Cas9 system for therapeutic gene editing of certain transcription regulators. DOI: 10.3390/genes11060704
Author Response
Pin Lyu and Baisong Lu present a quality and well-written review manuscript focused on new advances in using virus-like particles and related technologies for eukaryotic genome editing delivery.
Authors summarize recent developments in using virus-like particles for safe delivery of gene editing effectors. First they introduce some background information for easy understanding of how virus-like particles can be used for safe and efficient CRISPR/Cas9 delivery. Then they summarize recently developed virus-like particles as genome editing vehicles. Finally, they discuss their applications and future directions.
Authors cover such aspects as background knowledge for developing virus-like particles as genome editing delivery vehicle; recently developed virus-like particles as safe genome editing delivery vehicles; applications of virus-like particles in genome editing.
Finally, authors conclude that although virus-like particles have not been tested in clinical trials for genome editing, they have been successfully used in vitro in clinically relevant human cells and in vivo in disease mouse models. Authors hope that in the near future virus-like particles will become delivery vehicles for genome editing in clinical applications to benefit patients.
Overall, the manuscript is highly valuable for the scientific community and should be accepted for publication after the corrections are made.
Thanks for the positive comments.
Other comments:
1) Please check for typos throughout the manuscript.
A professional editor has edited our manuscript.
2) With regards to gene editing authors are kindly encouraged to cite the following article that describes the use of CRISPR/Cas9 system for therapeutic gene editing of certain transcription regulators. DOI: 10.3390/genes11060704
This reference is now cited as reference 32.

Reviewer 2 Report
The authors review recent advances in using the technique of virus-like particles (VLPs) for in vitro and in vivo delivery of gene editing effectors. Given the wide application of genome editing approach in basic and applied research, the development of delivery methods is of general interest. The manuscript is well written and provides an updated analysis of VLPs-mediated delivery strategies. It may help to identify future directions in this filed. I have several minor issues that the authors may consider in a revised version.
1. In section 3, the authors present examples of VLPs-mediated methods. In some instances, it would be helpful to the readers if the authors could provide more detail on the edited genes. For example, in section 3.1.1.2 the authors describe that “Most interestingly, systemically delivered eVLPs achieved efficient base editing in mouse liver”. I am interested in knowing the genes edited and the effects produced.
2. Table 2 shows several in vitro and in vivo delivery experiments. Similarly, it would be better to further precise the targeted genes and tissues. The authors may consider adding a column for this.
3. In figure 2B, it is unclear for me whether proteins and RNPs packaged through the fusion strategy are also delivered by using exosomes as vehicles. By the way, the authors need to clarify whether exosome packed with gene editing effectors are considered as VLPs or as an alternative to VLPs.
4. In the Summary section, the authors discuss advantages and limitations for different delivery strategies. The manuscript can be strengthened if the authors could propose future directions that help to further improve the delivery strategies.
Author Response
The authors review recent advances in using the technique of virus-like particles (VLPs) for in vitro and in vivo delivery of gene editing effectors. Given the wide application of genome editing approach in basic and applied research, the development of delivery methods is of general interest. The manuscript is well written and provides an updated analysis of VLPs-mediated delivery strategies. It may help to identify future directions in this filed. I have several minor issues that the authors may consider in a revised version.
- In section 3, the authors present examples of VLPs-mediated methods. In some instances, it would be helpful to the readers if the authors could provide more detail on the edited genes. For example, in section 3.1.1.2 the authors describe that “Most interestingly, systemically delivered eVLPs achieved efficient base editing in mouse liver”. I am interested in knowing the genes edited and the effects produced.
We have included more information about the genes edited and the effects produced (The last sentence of section 3.1.1.2).
- Table 2 shows several in vitro and in vivo delivery experiments. Similarly, it would be better to further precise the targeted genes and tissues. The authors may consider adding a column for this.
A column is added in Table 2 to provide the suggested information.
- In figure 2B, it is unclear for me whether proteins and RNPs packaged through the fusion strategy are also delivered by using exosomes as vehicles. By the way, the authors need to clarify whether exosome packed with gene editing effectors are considered as VLPs or as an alternative to VLPs.
In the fusion strategy, the cells will most likely also produce exosomes. However, since there is no mechanism to enrich RNPs into the exosomes, the genome editing activity of the exosomes is expected to be minimal. The exosomes are not regarded as VLPs, we discussed exosomes in this review since exosomes have similar sizes and membranes as VLPs. We have clarified this in the revision.
- In the Summary section, the authors discuss advantages and limitations for different delivery strategies. The manuscript can be strengthened if the authors could propose future directions that help to further improve the delivery strategies.
We have added such discussion in the revision.
